

# Simulated dynamic regrounding during marine ice sheet retreat

Lenneke M. Jong[1,2], Rupert M. Gladstone[1,3], Benjamin K. Galton-Fenzi[1,4], and Matt A. King[5]

[1]Antarctic Climate & Ecosystems Cooperative Research Centre, University of Tasmania, Private Bag 80, Hobart, Tasmania 7001, Australia
[2]Institute for Marine and Antarctic Studies, University of Tasmania, Private Bag 129, Hobart, Tasmania 7001, Australia
[3]Arctic Centre, University of Lapland, P.O. Box 122, 96101, Rovaniemi, Finland
[4]Australian Antarctic Division, Channel Highway, Kingston, Tasmania 7050, Australia
[5]School of Land and Food, University of Tasmania, Hobart 7001, Australia

*Correspondence to:* L. M. Jong (lenneke.jong@utas.edu.au)

**Abstract.**

Marine terminating ice sheets are of interest due to their potential instability, making them vulnerable to rapid retreat. Modelling the evolution of glaciers and ice streams in such regions is key to understanding their possible contribution to sea level rise. The friction caused by the sliding of ice over bedrock, and the resultant shear stress, are important factors in determining the velocity of sliding ice. Many models use simple power-law expressions for the relationship between the basal shear stress and ice velocity or introduce an effective pressure dependence into the sliding relation in an ad hoc. manner. Sliding relations based on water-filled sub-glacial cavities are more physically motivated, with the overburden pressure of the ice included. Here we show that using a cavitation based sliding relation allows for the temporary regrounding of an ice shelf at a point downstream of the main grounding line of a marine ice sheet undergoing retreat across a retrograde bedrock slope. This suggests that the choice of sliding relation is especially important when modelling grounding line behaviour of regions where potential ice rises and pinning points are present and regrounding could occur.

## 1 Introduction

Marine ice sheets, which are grounded below sea level, have been identified as having the potential to contribute significantly to future sea level rise through the rapid loss of ice under changing climate conditions. When grounded on a retrograde sloping bedrock (i.e. sloping downwards towards the interior of the continent) it has been suggested (Mercer, 1978; Weertman, 1974) that the positive relationship between ice thickness and ice flux leads to a positive feedback in which rapid retreat of the grounding line may occur, termed "marine ice sheet instability" (MISI). MISI theory predicts that the grounding line of a glacier cannot stabilise on a retrograde bedrock slope. Thus, if a glacier retreats onto a region with such a bedrock geometry it will continue retreating rapidly at least until reaching a prograde slope, potentially discharging large amounts of previously grounded ice into the ocean. Large regions of the Antarctic ice sheet, particularly in West Antarctica, are grounded below sea level with retrograde sloping bedrock (Fretwell et al., 2013) and thus may be susceptible to MISI. Modelling of the dynamics of marine ice sheets have been investigated widely, see Pattyn et al. (2017) for a recent review of the field, and have been the subject of recent model intercomparison projects such as MISMIP (Pattyn et al., 2012), MISMIP3d (Pattyn et al., 2013)



and MISMIP+ (Asay-Davis et al., 2016) and when focusing on retreating glaciers such as Pine Island (Gladstone et al., 2012; Joughin et al., 2010; Favier et al., 2014). More recent analysis has shown that stable grounding line configurations may be possible on retrograde sloping bedrock when the buttressing of floating ice shelves and 3D geometry of the system is included (Katz and Worster, 2010; Gudmundsson et al., 2012; Gudmundsson, 2013).

Many ice sheet models use a power-law relationship between the basal shear stress and sliding velocity, such that of Weertman (1957). These models require very fine mesh resolution in the vicinity of the grounding line (Vieli and Payne, 2005; Gladstone et al., 2010; Cornford et al., 2013). The sharp change in shear stress across the grounding line leads to difficulties in achieving a convergent result without very fine mesh resolution (Gladstone et al., 2017). Other relations have been investigated which take into account the effective pressure of the ice at its base either determined empirically such as that of Budd et al.

(1984), or by including in the presence of water-filled cavities (Schoof, 2005). Sliding relations in ice sheet models which include an effective pressure dependence, such as that implemented in Elmer/Ice (Gagliardini et al., 2007), provide a theoretically based treatment of basal friction, do not cause such strong mesh resolution dependency and do not suffer from the same mesh resolution issues as Weertman type relations and satisfy Iken's bound (Iken, 1981). Recently Tsai et al. (2015) explored the effect of using a modification to the power-law basal sliding with Coulomb friction used close to the grounding line on the

stability and profiles of marine ice sheets.

    The underlying geometry of the bedrock is an important control in the stability of marine ice sheets. The role of ice rises and pinning points (Favier et al., 2012; Favier and Pattyn, 2015; Fürst et al., 2016) in affecting buttressing forces and stabilisation of the grounding line has been investigated in numerical modelling efforts, as has the role of glacial isostatic adjustment (Gomez et al., 2012). Recent ensemble simulations of Antarctic Ice Sheet deglaciation since the last glacial maximum (Kingslake et al.,

2017) have demonstrated that regrounding of pinning points in large ice shelves due to glacial uplift after a period of retreat can cause a stabilisation and readvance of the grounding line. Thus regrounding can be important to large scale marine ice sheet dynamics, even leading to a partial recovery from MISI.

    In this study, we further investigate the impact of the sliding law on glacier trajectory in an idealised 2D flowline model, showing that dynamic regrounding on a retrograde bedrock slope can occur when a sliding relation with a dependency on the

effective pressure at the base of the ice is used.

## 2   Methods

### 2.1   Model description

In this study we use a finite element model Elmer/Ice (Gagliardini et al., 2013) to solve the full Stokes equations for a viscous fluid. A rheology following Glen's law is used, with the temperature held constant through the whole of the ice sheet at -15 C.

We use a two-dimensional flowline geometry with the bedrock shape the same as that used in a recent model intercomparison project (Pattyn et al., 2012), featuring a section of retrograde slope, described by





$$B(x) = 729 - 2184.8\left(\frac{x}{750\text{km}}\right) + 1031.72\left(\frac{x}{750\text{km}}\right)^4$$
$$- 151.72\left(\frac{x}{750\text{km}}\right)^6, \tag{1}$$

where $x$ is distance from the inland boundary and $B$ is bedrock elevation relative to sea level.

The grounding line position is solved in the model through a contact problem, taking into account the geometry of the lower surface with respect to the bedrock, the effective pressure at the base of the grounded ice and the buoyancy of the ice in contact with the ocean.

We use a basal sliding relation that is based on the theory of sliding with cavitation (Schoof, 2005) and has been implemented in Elmer/Ice (Gagliardini et al., 2007). The basal friction is related to the sliding velocity by

$$\frac{\tau_b}{N} = C\left(\frac{\chi}{1+\alpha\chi^q}\right)^{1/n} \tag{2}$$

where $\tau_b$ is the basal shear stress, $u_b$ the basal ice sliding velocity

$$\chi = \frac{u_b}{C^m N^n A_s} \tag{3}$$

and

$$\alpha = \frac{(q-1)^{q-1}}{q^q}. \tag{4}$$

$A_s$ is the sliding parameter in the absence of cavitation , $n$ Glen's law exponent, $C$ the maximal value of $\tau_b$ and $q \geq 1$ the exponent controlling the post-peak decrease. The effective pressure $N$ is calculated based on the assumption of full connectivity between the subglacial hydrologic system and the ocean. This sliding relation is henceforth referred to as the "cavitation sliding relation".

The other sliding relation used in this study, is that non-linear Weertman-type friction law, of the form:

$$\tau_b = A_s.u_b{}^{m-1}.u_b \tag{5}$$

where $m$ in this case is $1/n$, $\tau_b$ and $u_b$ the basal shear stress and ice velocity respectively and $A_s$ is a constant sliding coefficient. This sliding relation is henceforth referred to as the "Weertman sliding relation".

Buttressing is included using a parameterisation of lateral drag by (Gagliardini et al., 2010) where the lateral resistance parameter $K$ is given by:

$$K = \eta \frac{(n+1)^{\frac{1}{n}}}{(\rho_{ice} W^{1+\frac{1}{n}})} \tag{6}$$

with $\eta$ the effective viscosity of the ice, $\rho_{ice}$ the density of the ice and $n = 1/m$. $W$ is a parameter corresponding to a channel width which we use to modify the lateral drag from high to low in order to force the glacier to retreat.



The experiments presented here used a horizontally uniform mesh resolution with a 500m element size for simulations using the cavitation sliding relation, and 250m element size for simulations using the Weertman sliding relation. Tests were carried out at coarser resolutions to test for resolution dependence (see Appendix A for details and B for model parameter values). The mesh is extruded in the vertical direction to 20 equally spaced layers in all simulations.

## 2.2   Experimental Description

The retreat experiments examine the behaviour of an ice sheet retreating across a section of retrograde slope when using the cavitation sliding relation, which incorporates a dependency of basal shear stress on effective pressure at the base of the ice. We begin with a 2D ice sheet grown from a uniform initial thickness of 100m. During the initial 5000 years of spinup the parameterised buttressing is set to zero. The buttressing is then linearly increased from zero to an effective channel width of 100km over the next 5000 years, resulting in a high buttressing force. The model is then continued until the grounding line position rests on the seaward side of the bedrock overdeepening (at approximately 1400km from the ice divide). During this initial spinup period the top surface accumulation rate is set to $1\mathrm{m\ yr}^{-1}$, to reduce the total run time for the spinup. After 16600 years the accumulation rate is then decreased to $0.3\mathrm{m\ yr}^{-1}$ and held at this value while the ice sheet stabilises and remains at this value throughout the retreat experiments. We determine that the spinup has finished and the ice sheet has reached a steady state when there has been no change in the grounding line position, and the mesh velocity, determining the change in the top and bottom free surfaces, remains less than $0.001\mathrm{m\ yr}^{-1}$ over 10000 years, resulting in a total spinup time of 25000 years.

We run a series of experiments where we trigger retreat of the glacier through a reduction in the buttressing force by linearly increasing the channel width parameter over 10 years to reduce the buttressing force, using values of $W$ equal to 250km, 300km, 350km, 375km, 400km and 450km. An infinitely wide channel corresponds to the case of no lateral drag an the values of $W$ used in the experiments should be considered as simply providing a range of values for buttressing. Simulations are then run for 2500 years, with 0.1 year timesteps with no further forcing change applied after the initial buttressing adjustment.

We carry out a similar retreat experiment using the Weertman sliding relation (equation 5). In this experiment we again spin up the ice sheet, initially for 5000 years with no buttressing and accumulation of $1\mathrm{m\ yr}^{-1}$, increasing the buttressing with $W = 100$km linearly over 5000 years. The accumulation rate is then reduced to $0.3\mathrm{m\ yr}^{-1}$ and the model is left to evolve for a further 10000 years until the top and bottom free surfaces show minimal change, resulting in a total spinup time of 20000 years. We chose a spatially uniform value for the Weertman sliding friction coefficient to match the value of basal shear stress observed in the cavitation sliding relation experiments at maximum values of ice velocity and height. Again, we trigger a retreat of the grounding line by reducing the buttressing, increasing $W$ from 100km to 350km linearly over 10 years and then continuing to let the simulation run without further forcing. A timestep of 0.5 years was used throughout the run.



## 3 Results

### 3.1 Cavitation sliding

The position of the grounding line is tracked over time during the channel width increase and through the continuation of
the model run (Figure 1a). In most simulations using the cavitation sliding relation the grounding line has retreated across

the retrograde slope within the first 1000 years, and by 2500 years grounding lines and surface slopes are stabilising. The
simulations with higher buttressing (i.e. a smaller forcing perturbation relative to the pre-retreat state) take longer to retreat and
stabilise.

The simulations with $W = 350, 375$km feature a temporary regrounding of the ice shelf during retreat on the retrograde
bedrock. The temporary regrounding occurs approximately 200km downstream of the original (henceforth "upstream") ground-

ing line. The upstream grounding line continues retreat during regrounding of the shelf.

This regrounding is likely caused by the downstream advection of thicker interior ice, which is mobilised by the buttressing
reduction. In general, dynamic thinning of the ice shelf due to reduced buttressing competes with thickening due to downstream
advection of thicker interior ice to give either a net thinning or thickening in the shelf. For simulations with parameterised
channel width $W \geq 400$km the peak ice discharge comes early (approximately 300 years, see also Figure 1b) and dynamic

thinning is sufficient to prevent regrounding. For simulations with parameterised channel width $W \leq 300$km a sharp peak in
discharge is not seen (Figure 1b), and downstream advection of thicker ice is slow, too slow to overcome dynamic thinning in
the shelf. Thus the simulations with $W = 350, 375$km represent a key region of input space in which regrounding may occur.
We refer to this as "dynamic regrounding" to distinguish it from regrounding due to bedrock uplift (described in Section 1).

The position of the flux gate in Figure 1b (1200km from the ice divide) is chosen as it is located where the regrounding

occurs. The flux reaches a maximum as the grounding line approaches the inland end of the retrograde bedrock region, and
decreases as the grounding line migrates up the prograde slope. For the cases where dynamic regrounding occurs we see a
temporary reduction in the flux, but this reduction is not sufficient to stabilise retreat.

Figure 2 presents a more detailed analysis of retreat and dynamic regrounding for $W = 350$km. It can be seen that the slope
of the lower surface of the ice shelf is similar to the retrograde bedrock slope, and this corresponds to a very shallow water

column under the shelf. The implication is that only a very small amount of thickening is needed to cause regrounding. The
shallow water column is common to all retreat simulations with the cavitation sliding relation (not shown).

Figure 2c shows basal shear stress, the point of inflection, and the grounding line. The point of inflection is with respect
to the upper surface height of the ice sheet, and indicates the switch from a convex ice sheet surface (inland, which includes
most of the grounded ice) to concave. It is identified here through calculation of the maximum gradient of the upper surface.

Figure 2c shows that the point of inflection corresponds to the maximum in basal shear stress. Downstream of this line basal
shear stress drops rapidly to zero, due to the dependence on effective pressure at the bed, $N$, in the cavitation sliding relation
(equation 2). The grounding line is persistently a few tens of km downstream of the point of inflection. The implications of the
point of inflection and basal shear stress pattern for transition zones and ice sheet profiles with different sliding relations will



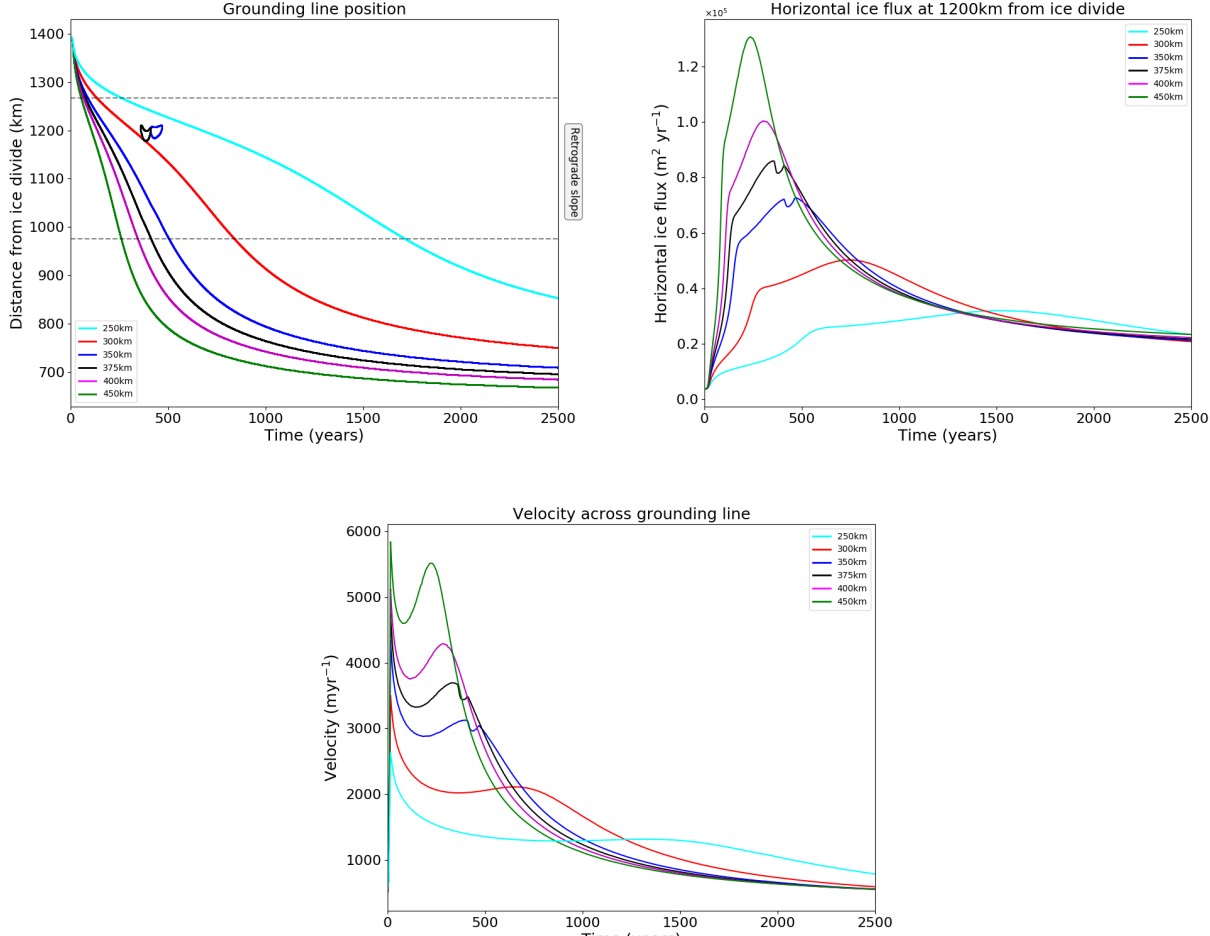

**Figure 1.** Retreat experiments using the cavitation sliding relation. (a) Evolution of grounding line position over time for a range of parameterised channel widths. Dashed lines indicate the extent of the retrograde bedrock. (b) Total flux of ice in the horizontal direction across a line 1200km from the ice divide. (c) Sliding speed across the grounding line.

be discussed further in Section 4. Basal shear stress is also very low under the regrounding region, again due to the dependence on $N$.

## 3.2 Weertman sliding

In experiments where the Weertman sliding law is used we see no temporary regrounding of the ice sheet. The ice shelf
5    develops a thinner profile than with the cavitation sliding relation and a markedly different shape, as shown in Fig. 3a. A
strongly concave shape is evident immediately downstream of the grounding line, in contrast to the more linear ice shelf profile



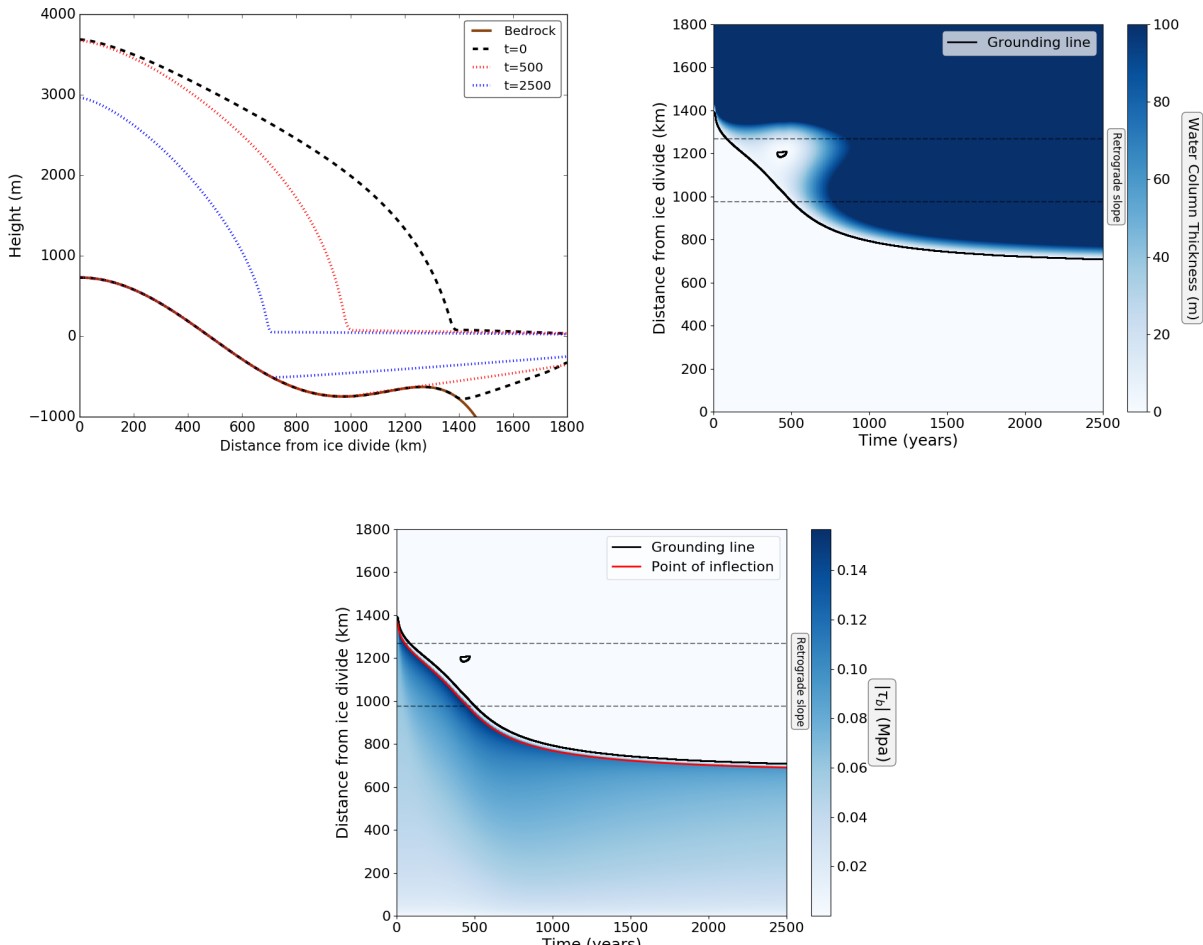

**Figure 2.** Retreat experiment details with the cavitation sliding relation and $W = 350$km. (a) Ice sheet profiles during retreat. Bedrock profile is also shown. (b) Evolution of the water column thickness during retreat. (c) Evolution of the basal shear stress.

when using the cavitation sliding relation. As a result of this difference, water column thickness is much larger during retreat than for the cavitation sliding relation.

## 4 Discussion

Retreat simulations in the current study have demonstrated that regrounding of an ice shelf may occur under certain conditions
during the rapid, unstable retreat of a marine ice sheet. In our simulations, this regrounding has a modest negative impact on
the rate of mass loss from the grounded ice sheet, but we cannot rule out the possibility that such behaviour could stabilise a



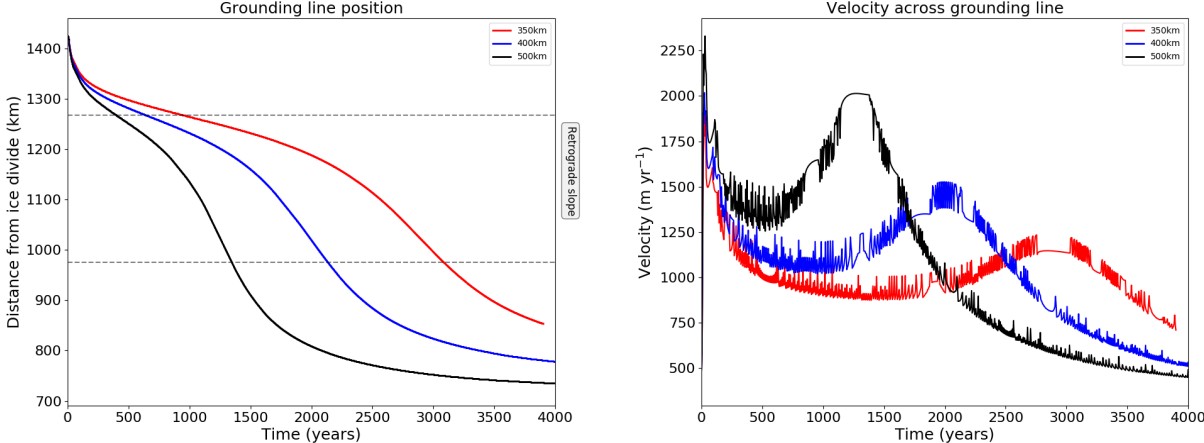

**Figure 3.** Evolution of (a) grounding line position and (b) cross-grounding line sliding speed for a range of parameterised channel widths using the Weertman sliding relation.

retreating ice sheet for certain geometries. The regrounding occurs as thick, previously grounded ice is advected downstream toward a bedrock rise in response to reduced buttressing.

While a thorough investigation into conditions for regrounding to occur is beyond the scope of the current study, it does provide insight into possible conditions when regrounding may occur. The geometry of the ice sheet is clearly important, as

advection of thicker ice is required to cause regrounding. This may be more complicated in three dimensions - flow convergence may itself provide thickening in response to grounded ice speed up. Bedrock geometry is also important - the retrograde slope is a key feature of the current setup and it is unlikely that regrounding could occur without an overdeepening. We suggest that a higher (closer to sea level) bedrock maximum and steeper retrograde slope are both likely conducive to regrounding. Choice of sliding relation is also important. The strongly concave lower surface of the ice shelf ice shelf just downstream

from the grounding line in the case of Weertman sliding increases the water column depth under the ice shelf and reduces the likelihood of regrounding. High sub-shelf melt rates may also cause a concave lower surface profile, reducing the likelihood of regrounding.

Impact of sliding law merits further consideration. The force balance in the ice sheet changes from the grounded region to the floating ice shelf over what is typically termed a "transition zone" (Pattyn et al., 2006; Schoof, 2007a) . Grounded ice

typically features high gravitational driving stress and high basal shear stress, especially for the high surface gradients and high velocities as the transition zone is approached. The floating ice shelf features smaller magnitude forces with the parameterised buttressing approximately balancing longitudinal stress, and driving stress being close to zero. In the absence of buttressing the ice shelf thickness would be almost constant through most of its length (Schoof, 2007b). For the case of Weertman sliding, the instantaneous decay of basal shear stress from its maximum value to zero as the grounding line is crossed (Figure 3) causes

the transition zone to extend into the iceshelf several km (typically around 20km in our experiments) causing the concave





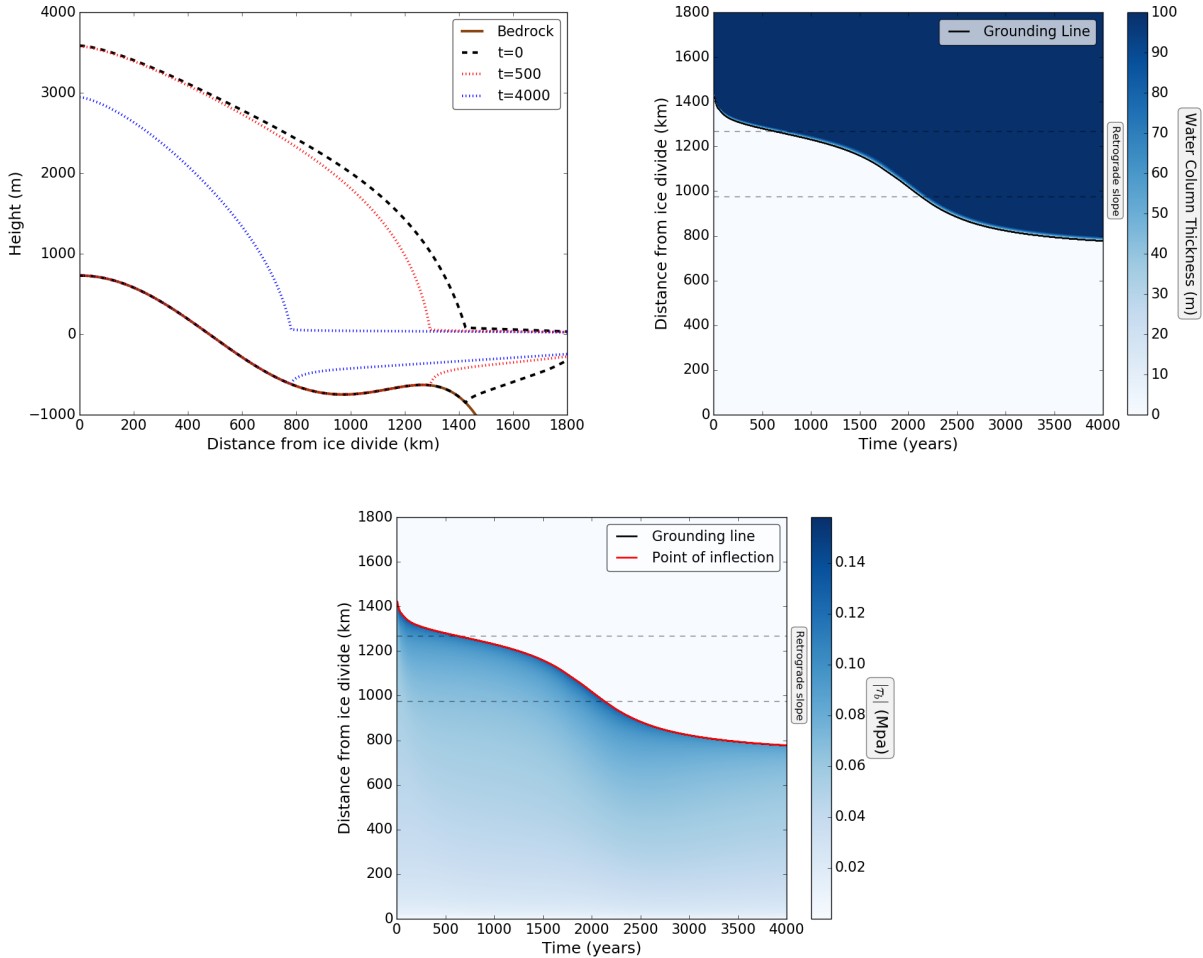

**Figure 4.** Retreat experiment details with the Weertman sliding relation and $W = 400$km. (a) Ice sheet profiles during retreat. Bedrock profile is also shown. (b) Evolution of the water column thickness during retreat. (c) Evolution of the basal shear stress. Note that the point of inflection and grounding line position are co-located.

and rapidly thinning iceshelf in this region. For the cavitation sliding relation, the rapid decay of basal shear stress to zero in the vicinity of the point of inflection in surface slope (Figure 2) leads to a grounded transition zone with the concave ice thickness occurring upstream of the grounding line and resulting in a thicker, more linear, ice shelf. Figure 5 shows example ice sheet/shelf geometries over the transition zone for both sliding relations. This difference in ice shelf profile is a direct result of dependence on effective pressure at the bed and is likely to be present also for other sliding relations featuring such a dependence, e.g. Budd et al. (1984).




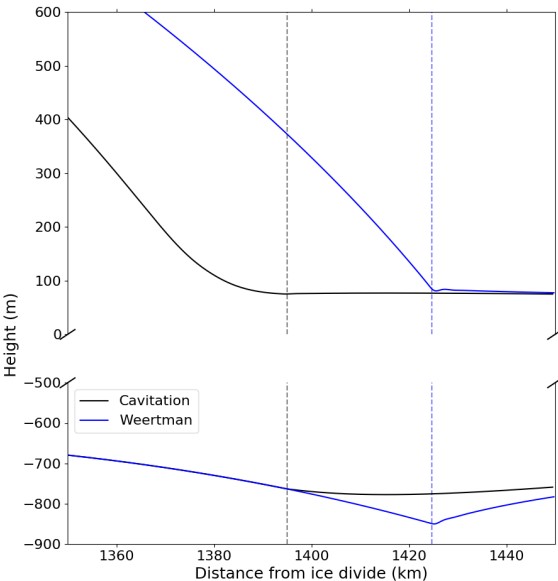

**Figure 5.** Detail of ice sheet profiles at t=0 for both cavitation sliding and Weertman sliding. Dashed vertical lines indicate the position of the grounding line.

The contrasting ice shelf profiles for the two sliding relations in the current study could indicate a means for validation of the choice of sliding relation through comparison against observed ice shelf profiles. Ideally, observed profiles for simulations with low ice shelf basal melting and low buttressing would be used, as both of these factors could impact on the ice shelf geometry.

The dynamic regrounding in the current study has some similarity to the regrounding that occurs following the overshoot mechanism presented by Kingslake et al. (2017), in which regrounding of an ice shelf after retreat has stabilised may occur through bedrock uplift after ice unloading. Both regrounding mechanisms impose a reduction in ice discharge, though the overshoot regrounding is lasting and the dynamic regrounding in the current simulations is transient. The suggested timescale for overshoot regrounding is an order of magnitude greater than the timescale for dynamic regrounding, but these timescales are also controlled by the size of the system, the ice flow speeds, and potentially ice and bedrock geometry. In the case of overshoot regrounding the timescale is also strongly dependent on mantle viscosity. It may be possible that for some bedrock configurations, and in the case of low mantle viscosity, dynamic regrounding could sufficiently retard ice sheet retreat to prevent overshoot and allow a post-retreat steady state to be reached more quickly.



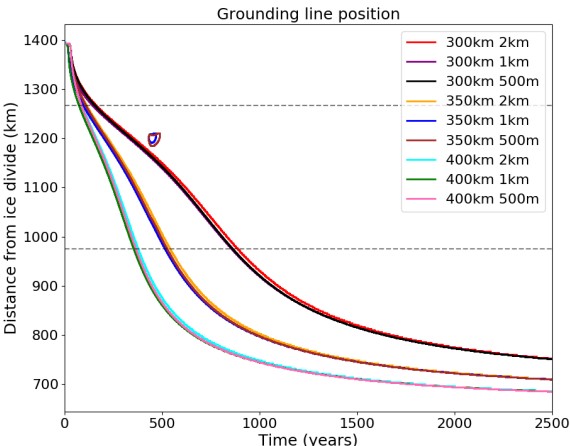

**Figure 6.** Grounding line retreat for the mesh resolution dependence experiments using the cavitation sliding relation. Buttressing parameters of 300km, 350km and 400km are used with 2km,1km and 500m uniform element size in the horizontal.

## 5   Conclusions

Flowline ice sheet simulations carried out in the current study demonstrate that regrounding on a retrograde bedrock slope can occur during marine ice sheet retreat. It is not yet clear under what conditions this regrounding, combined with isostatic rebound, could counter retreat and stabilise the ice sheet.

The current study also demonstrates that use of a sliding relation in which basal shear stress is dependent on effective pressure at bed impacts on transition zone location and on ice shelf thickness profiles immediately downstream of the grounding line. This dependence implies that regrounding is less likely to occur when a Weertman sliding relation is used, and could provide a means for validating choice of sliding relation through comparison with observed ice shelf profiles.

### Appendix A:  Mesh Resolution Dependence

A number of experiments were run at different mesh resolutions demonstrating convergent behaviour. Identical retreat experiments were performed using the cavitation sliding relation on uniform meshes with 2km, 1km and 500m grid spacing. Runtime is considerably longer for the finest mesh resolution. The results, shown in Fig. 6, demonstrate that the regrounding behaviour is present and almost identical in both the 1km and 500m mesh spacing cases. For each channel width, the final position of the grounding line is the same for all mesh resolutions. We conclude that the effects shown here are not dependent on the mesh

resolution and a finer resolution is not required to get convergent behaviour of the grounding line.

Similar experiments were preformed using the Weertman sliding relation on uniform meshes with 1km, 500m and 250m grid spacing. In this case, while the end position of the grounding line appears to converge, the transient behaviour of the





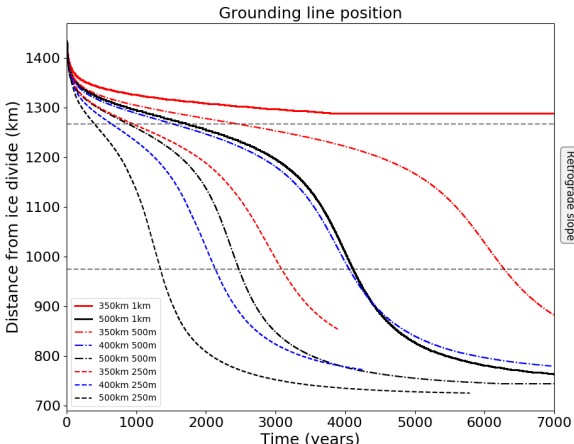

**Figure 7.** Grounding line retreat for the mesh resolution dependence experiments using the Weertman sliding relation. Buttressing parameters of 350km, 400km and 500km are used with 1km,500m and 250m uniform element size in the horizontal.

grounding line is not consistent between the grid sizes, with the finest mesh resulting in the fastest retreat of the grounding line across the retrograde slope. The concave geometry of the lower surface of the ice shelf is consistent across the mesh resolution experiments. Previous studies into the mesh resolution dependence of the grounding line using Weertman sliding have demonstrated convergence of the grounding line position, eg. (Durand et al., 2009).

5 **Appendix B: Model Parameters**

*Acknowledgements.* The authors would like to thank Thomas Zwinger for technical help with the simulations. Rupert Gladstone is funded by the Academy of Finland, grant number 286587. This research was undertaken with the assistance of resources from the National Computational Infrastructure (NCI), which is supported by the Australian Government. This work was supported in part by the Australian Government's Cooperative Research Centres Program through the Antarctic Climate and Ecosystems Cooperative Research Centre (ACE CRC).
10 This research was supported under Australian Research Council's Special Research Initiative for Antarctic Gateway Partnership (Project ID SR140300001).



| Parameter | Value | Unit | Description |
|---|---|---|---|
| $n$ | 3 | | Glen's law exponent |
| $T$ | -15 | °C | ice temperature |
| $\rho_i$ | 910 | kg m$^{-3}$ | density of ice |
| $\rho_w$ | 1000 | kg m$^{-3}$ | density of water |
| $g$ | -9.8 | m s$^{-2}$ | Gravitational acceleration |
| $a$ | 0.3 | m yr$^{-1}$ | Accumulation rate |
| $C$ | 0.1 | | Cavitation friction law maximum value |
| $q$ | 1 | | Cavitation friction law post peak exponent |
| $m$ | 3 | | Power Law exponent (m=n= Glen's Exponent) |
| $A$ | $3 \times 10^{-25}$ | s$^{-1}$ Pa$^{-3}$ | Fluidity parameter |
| $A_s$ (Cavitation) | $4.1613 \times 10^5$ | Pa m$^{-1/3}$a$^{1/3}$ | Cavitation friction law sliding parameter |
| $u_{t0}$ | 0.01 | m yr$^{-1}$ | Cavitation friction law linear velocity |
| $A_s$ (Weertman) | $3.812 \times 10^6$ | Pa m$^{-1/3}$a$^{1/3}$ | Weertman sliding parameter |

**Table 1.** Values of model parameters used

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
