# Peer review of "Simulated dynamic regrounding during marine ice sheet retreat"

_The Cryosphere, 2017_

## Referee Comment (RC1) · J. Todd (Referee) · 8 Nov 2017

**Review of "Simulated dynamic regrounding during marine ice sheet retreat" by Jong et al.**

General Comments:

This study uses 2D flowline simulations in Elmer/Ice to investigate the impact of the choice of sliding law on ice sheet and grounding line dynamics during retreat over retrograde bedslopes, and discusses the process of 'dynamic regrounding', whereby a retreating ice sheet may reground on a pinning point due to advection of thicker ice, potentially stabilising the retreat. The authors find that regrounding for this particular model domain is possible with the cavitation sliding law, but not with the Weertman law. The authors also propose that the shape of real ice sheet transition zones may provide clues as to the true nature of sliding under ice sheets.

This is an interesting piece of work which makes a valuable contribution to the field, and so merits publication. However, I have some concerns about the experimental setup which should be addressed.

The main results presented for cavitation and weertman sliding are at 500m and 250m, respectively, but the authors have also run the cavitation model at 1 and 2km resolution, and the weertman at 500m, 1km. Why not present and compare the 500m resolution results for both models?

The simulations also use different timestep sizes. For cavitation sliding, dt = 0.1 years, but for Weertman, dt = 0.5 years. Why? Can you be sure that these simulations, which have different mesh resolution and timestep size, are comparable?

Retreat of the grounding line is forced by reducing the effective channel width. This doesn't seem to reflect any hypothesised real-world drivers of ice sheet collapse, except perhaps progressive weakening of shear margins? Are your results sensitive to the nature of the retreat forcing? Would it not be more realistic to alter the SMB or apply basal melting to the floating tongue?

Specific Comments:

The comparison of the Weertman-style sliding vs cavitation sliding seems to centre on the fact that Weertman sliding is tricky to implement in models, but this isn't the only shortcoming of sliding laws which neglect effective pressure, surely? Can the authors not also make the case that Weertman sliding is fundamentally unsuitable for systems with significant water pressure at the base?

I guess water pressure at the base is simply defined by sea level? This should be explicitly stated in the methods, I think. Is this what you mean by P3L15: "based on the assumption of full connectivity between the subglacial hydrologic system and the ocean"?

I think you could be clearer on how lateral drag is implemented. You describe the lateral resistance parameter K but don't explain how that modifies the Stokes solution. Also, when you say "… which we use to modify the lateral drag from high to low", do you mean that the channel width varies through space or time? This becomes clear later on, but it would be good to avoid confusion here.

Also, on the issue of lateral drag and channel width, the parameter W, as used by Gagliardini et al. (2010) refers to the half-width, rather than the width. Comparing your Eq. 6 with their supporting material, I see that you've adapted the equation

somewhat, so I am unsure whether this should *still* be half-width or if your adaptation accounts for this.

I like how you've presented the results in Figs. 2 and 4, but I found myself flicking back and forth between them for comparison. Perhaps you could reformat to show 2a alongside 4a, 2b alongside 4b, etc?

The velocity plots in Figure 3b are strange, and, I guess, indicative of mesh dependency? At any rate, the high frequency variability should be explained.

Technical Issues:

P2L1: "and when focusing on retreating glaciers such as Pine Island" Slightly odd wording?

P2L11: Elmer has a few sliding relations implemented, including Weertman and Budd.

Fig 1,2,3,4: Missing a,b,c labels

P3L14:  C is the max value of Tb/N, I believe.

P3L18: "is *the* non-linear Weertman-type"?

P3L20: Why not redefine *m* to something else to avoid confusion with *m* in Eq. 3

P3L22 and elsewhere: Perhaps "lateral buttressing" or "lateral drag" would be clearer? When I think of 'buttressing' I think of melange or sea-ice buttressing.

P4L28: Here you state that you increase W from 100km to 350km, but I think you also run 400 and 500km (Fig. 3), right?

P9L1: "iceshelf" => "ice shelf"

---

## Referee Comment (RC2) · Anonymous Referee #2 · 14 Dec 2017

**General comments**

'Simulated dynamic regrounding during marine ice sheet retreat' is a 2D flowline model study, which makes use of the finite element model Elmer/Ice to solve the full Stokes equations of ice flow dynamics in order to compare ice sheet retreat on a retrograde bedrock slope following buttressing reduction for two different friction laws, i.e. the friction law proposed by Schoof (Schoof, 2005) - which is called "cavitation sliding relation" in the manuscript - and the traditional Weertman friction law (Weertman, 1957). The authors find out that depending on the magnitude of the perturbation (i.e. the amount of buttressing reduction relatively to the pre-retreat state) it is possible to obtain temporary "dynamic regrounding" when the Schoof friction law is used whereas the Weertman sliding law produces no "dynamic regrounding" at all for all the tested perturbations. There are also some reflexions about the different surface ice sheet profiles induced by the two different laws and the fact that these different profiles could be compared to observations in order to discriminate one of the two laws as the most plausible.

Overall, the results highlighted in this study are rather interesting, especially regarding the "dynamic regrounding" behavior and this paper ought to be published, yet I have three major criticisms to formulate.

First of all, the influence of the chosen friction law on the grounding line dynamics for a 1HD ice sheet resting on the bedrock shape designed for the MISMIP intercomparison exercise Pattyn and others (2012) has been investigated in great details by Brondex and others (2017). Therefore, I do think that this paper should at least be cited in yours. In particular, Brondex and others (2017) obtain thoroughly different GL dynamics with, respectively, the "cavitation sliding relation" (Schoof friction law) and Weertman friction laws following a loss of buttressing starting from identical steady states so that observed differences can only be due to differences in the friction law formulation; in addition, they also draw some conclusions about the effect of different friction laws on surface ice sheet profiles which turn out to be in line with the findings of Tsai and others (2015) and Gladstone and others (2017).

This lead me to the second major criticism: by construction, the two initial states which are used as the starting points of the perturbation experiments that you lead are not equivalent; one is obtained with the "cavitation sliding relation" and leads to a steady GL located at  $x_G \sim 1400$  km (from Fig. 1a, but it would be better if this value was given in the text) whereas the other is obtained with the Weertman law and leads to a steady GL located at  $x_G \sim 1430$  km (from Fig. 3a, but here too it would be better if this value was directly given in the text).

On page 4 line 26, there is an explanation on how the "Weertman sliding friction coefficient" (by the way, I think this terminology is confusing: it is either a "sliding coefficient" or a "friction coefficient" and if you refer to the parameter  $A_S$  that you use in equation (5) as I think you do, then it is a friction coefficient as  $\tau_b$  increases when  $A_S$  increases): it is said that a spatially uniform value of this coefficient is chosen "to match the value of basal shear stress observed in the cavitation sliding relation experiments at maximum values of ice velocity and height". This sentence is not very clear to me but I can say for sure that with this procedure the steady basal shear stress that you get with the Weertman law at the end of the spinup time differs from the one you get with the law of Schoof (only a spatially varying Weertman friction coefficient can reproduce, with a Weertman law, a reference basal shear stress field initially obtained with a Schoof law, see Brondex and others (2017)).

As a consequence, it is difficult to attribute the differences that you get with the Weertman and Schoof friction laws to differences in the friction law formulations rather than to differences on the initial steady states used as starting points for the perturbation experiments. Therefore, I would suggest to emphasize, in the discussion section, on the fact that the differences observed between the results produced with the two laws could be due not only to different dependencies of  $\tau_b$  on N and  $u_b$  (depending on the chosen friction law) but also to differences on the two initial states built with the two laws.

Finally, the "dynamic regrounding" that you observe with the Schoof friction law seems mostly due to the fact that with this law the bottom ice shelf profile appears to be rather linear whereas the one obtained with the Weertman law exhibits a concave shape. Looking at appendix B, it appears that the value you have chosen for the parameter C, i.e. C = 0.1, is rather low (see for example Pimentel and others (2010), Pimentel and Flowers (2011), Hewitt (2013), Leguy and others (2014), Asay-Davis and others (2016), Brondex and others (2017)). I would suggest you to test the sensitivity of your results to the value of this parameter as I would expect that, for higher values of C, the bottom ice shelf geometry becomes closer to the concave profile obtained with the Weertman law, and maybe to such an extent that you do not get dynamic regrounding with the Schoof law anymore. Indeed, I do think that the geometry that you get with the Schoof law is highly sensitive to the value attributed to the C parameter in equation (2): if you think in terms of asymptotic behaviors with the Schoof law, you can see that far from the GL (where N is very high) the Schoof law is almost perfectly equivalent to a Weertman law with  $\tau_b \to A_S u_b^m$  (by

the way, the way you use  $A_S$  in equation (2) is not consistent with the way you use it in equation (5), see specific comments); in contrast, in a narrow region located right upstream the GL (where N is very low) the School law is almost perfectly equivalent to a Coulomb law with  $\tau_b \to CN$ ; the horizontal extension of this region depends on the value attributed to C with lower values leading to wider extensions and presumably to stronger differences in ice shelf profiles relative to the ones produced by a Weertman law (the consequences of having a Coulomb friction regime at the GL on the surface ice sheet profile were investigated by Tsai and others (2015) but I don't think there is any conclusions about the ice shelf profiles).

Beside these major points, I have also noted several inconsistencies, especially regarding the formulation of the two friction laws. These inconsistencies are summarized in the "specific comments" section.

**Specific comments**

From P1 L21 to P2 L1: I don't understand this sentence, is there a problem with the end of it: "and when focusing on retreating glaciers such as Pine Island"

P2 L4: Brondex and others (2017) have also shown that tuning the spatial distribution of the Weertman friction coefficient to match the basal shear stress produced by a Schoof law could also lead, under certain circumstances, to steady GL positions located on the retrograde slope without having to add any lateral stress.

P2 L6-7: I have the feeling that the two sentences about the need for a very fine mesh resolution at the GL are redundant.

P2 L12: "do not cause such strong mesh resolution dependency and do not suffer from the same mesh resolution issues as Weertman type relations"  $\rightarrow$  here too I think that the information is redundant. In addition I would suggest to cite the work of Leguy and others (2014).

P2 L22: "even leading to a partial recovery from MISI"  $\rightarrow$  it is not clear what is meant by "partial recovery" here, i.e. is it the grounding line stabilizing on the retrograde slope? Or advancing back to its initial position?

P3 L11: I think there is a mistake in Eq. (3): it should be a *n* instead of the *m* for the exponent of the parameter *C*. Otherwise, you need to define the parameter *m* for this equation (but it is defined for Eq. (5) as m = 1/n, whereas it should be *n* for Eq. (3)).

P3 L11 and L19: the way you use the parameter  $A_S$  in Eq. (3) is not consistent with the way you use it in Eq. (5). Indeed, a dimensional analysis of Eq. (3) reveals that in this equation  $A_S$  should be given in  $myr^{-1}Pa^{-n}$  (which does not correspond to the unit you give in Table 1 of Appendix B) while it should be given in  $Pam^{-1/n}yr^{1/n}$  in Eq. (5).

The mistake probably comes from the fact that your Eq. (5) is not equivalent to Eq. (13) of Gagliardini and others (2007): they have  $u_b = A_S \tau_b^n$  (in this case  $A_S$  ought to be called a sliding parameter as an increase of  $A_S$  is associated to an increase of  $u_b$ ) while you have  $\tau_b = A_S u_b^{1/n}$  (in this case,  $A_S$  should be called a friction coefficient as an increase of  $A_S$  induces an increase of  $\tau_b$ ). I would suggest that you read again Gagliardini and others (2007) and rewrite properly all these equations. It could also be worth it to justify the fact that you decided to take q = 1 in Eq. (4), or at least to explain what is the role of this parameter. You also need to be clearer on the terminology "sliding parameter/coefficient" and "friction parameter/coefficient": most of the time the notation  $A_S$  is used for a sliding coefficient while in Eq. (5) you need a friction coefficient. More broadly speaking, I would rather speak about "friction law" instead of "sliding law" (or "sliding relation") when  $\tau_b$  is given as a function of other parameters.

P3 L24: once again, the expression that you give for K does not correspond to the original one of Gagliardini and others (2010). In particular, I don't really understand why  $\rho_i$  appears in your expression. In addition, in the current state of your manuscript it is impossible for the reader to understand how this parameterisation of lateral drag is included in the global stress balance, which is not straightfoward as we are considering a 1HD ice sheet.

P4 L16: "resulting in a total spinup time of 25000 years"  $\rightarrow$  It seems to me that this is not consistent with the description of the spinup procedure given in the previous lines: 16600 years of spinup with 1 m yr-1 of top surface

accumulation followed by, at least, 10000 years of spinup with 0.3 m yr-1 of top surface accumulation as you say that: "We determine that the spinup has finished and the ice sheet has reached a steady state when there has been no change in the grounding line position, and the mesh velocity, determining the change in the top and bottom free surfaces, remains less than 0.001m yr-1 over 10000 years"

P4 L19: "an"  $\rightarrow$  typo

P4 L26: as already said, the way you choose the Weertman law friction (or sliding ?) parameter is not clear to me.

P4 L28: you have also run simulations for W = 400 km and W = 500 km based on Fig. 3.

P5 L11-18: in my opinion, it would make more sense to have this part in the discussion section. In addition, I think that the point discussed here could be better illustrated by a plot of the thickness rates of change  $\partial H/\partial t$  as a function of x at different times following buttressing release.

P5 L23-26: as already said, this result might be highly sensitive to the value attributed to the C parameter in Eq. (2). Therefore, I would suggest to run similar simulations with higher values of this parameter.

P6 Fig1: I think you never refer to Fig. 1c in the text, therefore I wonder if it is really relevant to have it here.

P6 L5: "Fig. 3a"  $\rightarrow$  I think you mean Fig. 4a

P7 L6 and P8 L1: "but we cannot rule out the possibility that such behaviour could stabilise a retreating ice sheet for certain geometries."  $\rightarrow$  This is a too strong statement considering the results that are presented in your manuscript for which the transient regrounding obtained with the Schoof law for W = 350 km and W = 375 km are far from preventing the GL to retreat over the retrograde slope.

P8 L9: "ice shelf ice shelf"  $\rightarrow$  typo

P8 L13: "force balance"  $\rightarrow$  In my opinion, "stress distribution" or "stress state" would be more appropriate

P8 L20: "transition zone"  $\rightarrow$  I don't agree with the use you make of the term "transition zone" in this case. In line with Pattyn and others (2006), I understand the "transition zone" as being the narrow region right upstream the GL over which  $\tau_b$  progressively vanishes. By construction, the Weertman law with a uniform friction coefficient leads to a discontinuity of  $\tau_b$  at the GL which is equivalent to say that the length of the transition zone is reduced to 0. There cannot be any "transition zone" within the ice shelf - as you seem to suggest - as  $\tau_b = 0$  wherever ice is floating.

P9 L5: this result was already highlighted in Tsai and others (2015), Gladstone and others (2017) and Brondex and others (2017) (the Budd law being investigated in the two latter), and therefore a citation of these studies would be welcome here.

P10 L5-6: "in which regrounding of an ice shelf after retreat has stabilised may occur through bedrock uplift after ice unloading"  $\rightarrow$  I don't understand the meaning of this sentence, is there a problem with it ?

P10 L7: to me, it is not very clear to what timescale you refer. Is it the duration of the regrounding ?

P11 L13: "For each channel width"  $\rightarrow$  This formulation is misleading as the cases W = 250 km, W = 375 km and W = 450 km do not seem to be tested in your sensitivity analysis.

P11 L14: "We conclude that the effects shown here are not dependent on the mesh resolution"  $\rightarrow$  here too the formulation is misleading: if I am correct (based on Fig. 6), the 2 km mesh spacing case shows no regrounding at all even for W = 350 km so it is not correct to state that your results are not dependent on the mesh resolution.

P12 L3: "Previous studies into the mesh resolution"  $\rightarrow$  I am not a native english speaker but this formulation sounds odd to me

P12 L4: "eg. (Durand et al., 2009)"  $\rightarrow$  (eg. Durand et al., 2009)

P13: you need to correct the friction law parameters after having rewritten the equations of P3. In addition, m should be 1/3 and not 3.

**References**

- Asay-Davis XS, Cornford SL, Galton-Fenzi BK, Gladstone RM, Gudmundsson GH, Holland DM, Holland PR and Martin DF (2016) Experimental design for three interrelated marine ice sheet and ocean model intercomparison projects: Mismip v. 3 (mismip+), isomip v. 2 (isomip+) and misomip v. 1 (misomip1). Geoscientific Model Development, 9(7), 2471
- Brondex J, Gagliardini O, Gillet-Chaulet F and Durand G (2017) Sensitivity of grounding line dynamics to the choice of the friction law. Journal of Glaciology, **63**(241), 854–866
- Gagliardini O, Cohen D, Råback P and Zwinger T (2007) Finite-element modeling of subglacial cavities and related friction law. Journal of Geophysical Research: Earth Surface (2003–2012), **112**(F2)
- Gagliardini O, Durand G, Zwinger T, Hindmarsh R and Le Meur E (2010) Coupling of ice-shelf melting and buttressing is a key process in ice-sheets dynamics. Geophysical Research Letters, **37**(14)
- Gladstone RM, Warner RC, Galton-Fenzi BK, Gagliardini O, Zwinger T and Greve R (2017) Marine ice sheet model performance depends on basal sliding physics and sub-shelf melting. The Cryosphere, **11**(11), **319–329**
- Hewitt I (2013) Seasonal changes in ice sheet motion due to melt water lubrication. Earth and Planetary Science Letters, **371**, 16–25
- Leguy G, Asay-Davis X and Lipscomb W (2014) Parameterization of basal friction near grounding lines in a onedimensional ice sheet model. The Cryosphere, 8(4), 1239–1259
- Pattyn F, Huyghe A, De Brabander S and De Smedt B (2006) Role of transition zones in marine ice sheet dynamics. Journal of Geophysical Research: Earth Surface, 111(F2)
- Pattyn F, Schoof C, Perichon L, Hindmarsh RCA, Bueler E, de Fleurian B, Durand G, Gagliardini O, Gladstone R, Goldberg D, Gudmundsson GH, Lee V, Nick FM, Payne AJ, Pollard D, Rybak O, Saito F and Vieli A (2012) Results of the marine ice sheet model intercomparison project, mismip. The Cryosphere Discussions, 6(1), 267–308 (doi: 10.5194/tcd-6-267-2012)
- Pimentel S and Flowers GE (2011) A numerical study of hydrologically driven glacier dynamics and subglacial flooding. In Proceedings of the Royal Society of London A: Mathematical, Physical and Engineering Sciences, volume 467, 537–558, The Royal Society
- Pimentel S, Flowers G and Schoof C (2010) A hydrologically coupled higher-order flow-band model of ice dynamics with a coulomb friction sliding law. Journal of Geophysical Research: Earth Surface, **115**(F4)
- Schoof C (2005) The effect of cavitation on glacier sliding. Proceedings of the Royal Society A: Mathematical, Physical and Engineering Science, **461**(2055), 609–627
- Tsai VC, Stewart AL and Thompson AF (2015) Marine ice-sheet profiles and stability under coulomb basal conditions. Journal of Glaciology, 61(226), 205–215

Weertman J (1957) On the sliding of glaciers. Journal of glaciology, 3(21), 33-38

---

## Author Comment (AC1) · 28 Mar 2018

**General comments from reviewers**

We would like to thank the two reviewers for the useful feedback provided which has greatly assisted in improving the clarity and quality of our manuscript. We will address the general comments of the two reviews in order and then the specific comments of each to follow, detailing changes made.

Reviewer 1, Dr Todd, notes concern about the different mesh resolutions and time steps used for the two different sliding models. In both these cases we looked to getting convergent and self-consistent behaviour and used a time step that would allow that. We would also like to draw attention to the mesh resolution tests performed, the results of which are shown in Appendix 1 of the manuscript. For both sliding laws we ran tests with different mesh resolutions. We found that for the "cavitation" sliding law the resulting trajectory of the grounding line was unchanged between the 1km and 500m mesh resolution cases, while the 2km case had shown different results. Hence we concluded that finer mesh spacing was not required to be confident in resolving the grounding line position. The need for fine mesh resolution for models with a retreating grounding line is documented in previous studies for the Weertman sliding case and does indeed converge [1]. The concave shape of the grounded transition zone for the Cavitation sliding law, versus the steep gradients at the start of the ice shelf for the Weertman case remains consistent throughout all the mesh resolution test, hence we are confident that the regrounding behaviour is in general more likely for the cavitation sliding law case.

Dr Todd also notes that the forcing using effective channel width may not reflect any hypothesised real-world drivers of ice sheet collapse. We agree that forcing by the altering the smb or applying basal melting would be necessary for more realistic models. However, our model in already highly idealised and aiming only to investigate qualitative differences in the behaviour of the ice sheet purely due to the choice of sliding relations used, without further complications from other drivers of change. Certainly such experiments would be of interest but would form the basis of a different study than is in the scope of what we have presented here.

Reviewer 2 has 3 main points of concern. Firstly, they have highlighted the recent publication by [2] on a similar topic, using the same ice sheet model code, Elmer/Ice, which should be referenced. At the time of our submission of our paper it was not yet published. It is clearly highly relevant to our paper and we have cited it in our revised manuscript where appropriate.

The second point of concern for reviewer 2 is with our method of spinning up the system and then performing experiments using spatially uniform parameters for the two friction laws. We agree that, as highlighted by the reviewer, the resulting profile and the regrounding behaviour after retreat is a consequence of both the initial state of the ice sheet after spinup and the choice of friction law. While other studies, such as [2] have tuned the parameters of the sliding relations they use so that the distribution of basal shear stress is identical across all experiments initially, we have chosen here to use parameter values which will result in key properties of the system to be similar in both cases. In this case it is the initial grounding line position that we aim to begin with in a similar position. While tuning for identical initial states will allow for a directly comparable quantification of future changes, that is not the aim of our paper. In this paper (and perhaps other similar papers to some extent as the reviewer

points out) the ice sheet profiles have a distinct qualitative nature determined by the physics of the sliding relation. Tuning one sliding relation to match the state given by a different sliding relation provides an initial state that is inconsistent with the physics of the sliding relation. Our method of obtaining the initial state tuning the parameters to with similar values for ke allows us to then compare modes of behaviour.

The final major concern of reviewer 2 is that the "dynamic regrounding" is sensitive to the value of C ($C_{max}$ in our revised notation). We agree that a full sensitivity study would be of interest, but such is clearly a very large undertaking and beyond the scope of this work. We don't see an argument why C should be especially important. It should also be noted that the reviewer is concerned that our value of C is very low. We note that a higher value of C would lead to thicker grounded ice, and that it is the downstream advection of this thicker ice that causes the regrounding. So it could be argued that regrounding would be more likely with higher values of C.

In addition to these general comments, we have addressed the specific comments from the reviewers below, including those regarding inconsistencies and confusion in the formulation of the sliding laws in the original manuscript noted by both reviewers. These are detailed below.

**Specific Comments from reviewers**

**Reviewer 1**

*The comparison of the Weertman-style sliding vs cavitation sliding seems to centre on the fact that Weertman sliding is tricky to implement in models, but this isn't the only shortcoming of sliding laws which neglect effective pressure, surely? Can the authors not also make the case that Weertman sliding is fundamentally unsuitable for systems with significant water pressure at the base?*
We agree that for systems with significant water pressure at the base the Weertman sliding relation is unsuitable to use and recent modelling work on marine ice sheets especially shows that the results can vary widely depending on the friction law used. Our aim in this paper is not to provide evidence on the validity (or not) of Weertman sliding. This comparison, however, aims to point out qualitative differences in terms of ice sheet geometry rather than to try to claim that one sliding relation is "better" than the other.

*I guess water pressure at the base is simply defined by sea level? This should be explicitly stated in the methods, I think. Is this what you mean by P3L15: based on the assumption of full connectivity between the subglacial hydrologic system and the ocean?*
Yes, the water pressure is indeed defined by sea level. The paragraph around P3L15 has been edited to reflect this explicitly and now reads "The effective pressure N is calculated based on the sea level and the assumption of full connectivity between the subglacial hydrologic system and the ocean"

*I think you could be clearer on how lateral drag is implemented. You describe the lateral resistance parameter K but don't explain how that modifies the Stokes solution. Also, when you say ... which we use to modify the lateral drag from*

*high to low, do you mean that the channel width varies through space or time? This becomes clear later on, but it would be good to avoid confusion here.*

Please see our response to the similar comment from reviewer 2 below where we improve our explanation of the implementation of the lateral drag and clarify that we are varying the drag in time, rather than space.

*Also, on the issue of lateral drag and channel width, the parameter W, as used by Gagliardini et al. (2010) refers to the half-width, rather than the width. Comparing your Eq. 6 with their supporting material, I see that you've adapted the equations somewhat, so I am unsure whether this should still be half-width or if your adaptation accounts for this.*

In our original manuscript we used an equation for K as it is calculated in the Elmer/Ice online documentation, which varies by some factors (including $\rho_i$) than what is given in the supplementary material of Gagliardini et al. (2010) which first introduces the implementation of the lateral drag function. For consistency, in the manuscript we will use the same form and W should be the half-width of the channel. We have corrected the manuscript throughout so that W now refers to a channel half-width.

*I like how you've presented the results in Figs. 2 and 4, but I found myself flicking back and forth between them for comparison. Perhaps you could reformat to show 2a alongside 4a, 2b alongside 4b, etc.?*

We appreciate the feedback on the presentation of the results but feel that we have ordered the plots in the best way possible to convey the information When the manuscript is compiled in two column-format these plots are meant to stack on top of each other so that the horizontal axes align and features in the plots at co-incident times are clearer to see.

*The velocity plots in Figure 3b are strange, and, I guess, indicative of mesh dependency? At any rate, the high frequency variability should be explained.*

The amplitude of these high frequency fluctuations is expected to be a function of mesh resolution (amongst other things). Each jump in velocity probably corresponds to grounding line movement retreat to the next element. Ungrounding of an element significantly reduces the basal friction, allowing speed-up This does not happen with the Cavitation sliding law because the dependence on effective pressure means that the basal shear stress is in any case close to zero for elements that are newly ungrounded. We have added this explanation in a paragraph added in section 3.2 referring to this velocity plot.

*P2L1: and when focusing on retreating glaciers such as Pine Island Slightly odd wording?*

Yes, this paragraph has been edited and now reads: "Marine ice sheets have been investigated widely with Pattyn (2017b) providing a recent review of development in modelling their dynamics. They have been the subject of recent model intercomparison projects such as MISMIP (Pattyn 2012), MISMIP3d (Pattyn 2013) and MISMIP+ (Asay-Davis 2016) looking at idealised systems. Retreating glaciers showing a geometry making them susceptible to MISI, such as Pine Island Glacier, have been also been a particular focus (Gladstone 2012,Joughin 2010,Favier 2014). More recent analysis has shown that stable grounding line

configurations may be possible on retrograde sloping bedrock when the buttressing of floating ice shelves and 3D geometry of the system is included (Katz 2010,Gudmundsson 2012,Gudmundsson 2013).

*P2L11: Elmer has a few sliding relations implemented, including Weertman and Budd.*
We have added references to Gladstone et al. (2017) and Brondex et al. (2017) who have used the other sliding laws implemented in Elmer/Ice which include an effective pressure dependence.

*Fig 1,2,3,4: Missing a,b,c labels*
These labels have now been added on all these figures.

*P3L14: C is the max value of Tb/N, I believe.*
This has been corrected in the manuscript.

*P3L18: is the non-linear Weertman-type?*
This line now reads "is a non-linear, Weertman type friction law"

*P3L20: Why not redefine m to something else to avoid confusion with m in Eq. 3*
As noted below by reviewer 2, the m in Eq. 3 is an error and should in fact be n, so there is no longer with the m in Eq. 5 any confusion. The words "in this case" have been removed.

*P3L22 and elsewhere: Perhaps lateral buttressing or lateral drag would be clearer? When I think of 'buttressing' I think of melange or sea-ice buttressing.*
We have clarified this by altering the text to read "A buttressing-like force due to friction along glacier side walls is included by using a parametrisation of lateral friction by Gagliardini et al. (2010)".

*P4L28: Here you state that you increase W from 100km to 350km, but I think you also run 400 and 500km (Fig. 3), right?*
Yes, the text is amended to read "from 100km to values equal to 350km, 400km and 500km "

*P9L1: "iceshelf" → "ice shelf"*
This has been corrected in the manuscript.

**Reviewer 2**

*From P1 L21 to P2 L1: I don't understand this sentence, is there a problem with the end of it: "and when focusing on retreating glaciers such as Pine Island"* We agree this sentence was unclear, please see response above to a similar comment by the other reviewer.

*P2 L4: Brondex and others (2017) have also shown that tuning the spatial distribution of the Weertman friction coefficient to match the basal shear stress produced by a Schoof law could also lead, under certain circumstances, to steady GL positions located on the retrograde slope without having to add any lateral stress.*

We have added modified this sentence to include the reference to Brondex (2017) here. It now reads: "More recent analysis has shown that stable grounding line configurations may be possible on retrograde sloping bedrock when the buttressing of floating ice shelves and 3D geometry of the system is included (Katz and Worster, 2010; Gudmundsson et al., 2012; Gudmundsson, 2013) or in some configurations where the basal friction coefficient is tuned spatially (Brondex et al. 2017)."

*P2 L6-7: I have the feeling that the two sentences about the need for a very fine mesh resolution at the GL are redundant.*

We have combined these sentences to read as " These models require very fine mesh resolution due to the sharp change in sheer stress across the grounding line (Vieli et al. 2005, Gladstone et al. 2010, Cornford et al. 2013, Leguy et al. 2014, Gladstone et al. 2017).

*P2 L12: "do not cause such strong mesh resolution dependency and do not suffer from the same mesh resolution issues as Weertman type relations" → here too I think that the information is redundant. In addition I would suggest to cite the work of Leguy and others (2014).*

"and do not suffer from the same mesh resolution issues as Weertman type relations" has been deleted.

*P2 L22: "even leading to a partial recovery from MISI" → it is not clear what is meant by "partial recovery" here, i.e. is it the grounding line stabilizing on the retrograde slope ? Or advancing back to its initial position ?*

I think in this context a "partial recovery" simply means the grounding line advancing again after a period of retreat. It doesn't have to advance to its initial (pre-retreat) position in order to constitute a "partial recovery". We have modified this sentence to "even leading to a partial recovery from MISI with the grounding line advancing again after a period of retreat."

*P3 L11: I think there is a mistake in Eq. (3): it should be a n instead of the m for the exponent of the parameter C . Otherwise, you need to define the parameter m for this equation (but it is defined for Eq. (5) as $m = 1/n$ , whereas it should be n for Eq. (3)).*

This was a mistake, and Equation 3 is now corrected using n for the exponent of $C_{\max}$.

*P3 L11 and L19: the way you use the parameter $A_S$ in Eq. (3) is not consistent with the way you use it in Eq. (5). Indeed, a dimensional analysis of Eq. (3) reveals that in this equation $A_S$ should be given in myr 1 Pa n (which does not correspond to the unit you give in Table 1 of Appendix B) while it should be given in Pam 1/n yr 1/n in Eq. (5). The mistake probably comes from the fact that your Eq. (5) is not equivalent to Eq. (13) of Gagliardini*

*and others (2007): they have $u_b = A_S \tau_b$ (in this case $A_S$ ought to be called a sliding parameter as an increase of $A_S$ is $1/n$ associated to an increase of u b ) while you have $\tau_b = A_S u_b$ (in this case, $A_S$ should be called a friction coefficient as an increase of $A_S$ induces an increase of $\tau_b$ ). I would suggest that you read again Gagliardini and others (2007) and rewrite properly all these equations. It could also be worth it to justify the fact that you decided to take $q = 1$ in Eq. (4), or at least to explain what is the role of this parameter. You also need to be clearer on the terminology "sliding parameter/coefficient" and "friction parameter/coefficient": most of the time the notation A S is used for a sliding coefficient while in Eq. (5) you need a friction coefficient. More broadly speaking, I would rather speak about "friction law" instead of "sliding law" (or "sliding relation") when $\tau_b$ is given as a function of other parameters.*

We agree with the review that the notation and language used originally to describe the sliding relations and parameters was inconsistent and has caused confusion. We have decided to follow Brondex et al. (2017) using $C_{W,S,B}$ to differentiate between the friction coefficients in the different sliding relations. We have chosen to use "sliding relation" consistently in the manuscript.

*P3 L24: once again, the expression that you give for K does not correspond to the original one of Gagliardini and others (2010). In particular, I don't really understand why $_i$ appears in your expression. In addition, in the current state of your manuscript it is impossible for the reader to understand how this parametrisation of lateral drag is included in the global stress balance, which is not straightforward as we are considering a 1HD ice sheet.*

The lateral drag formula as it originally appears in our paper is consistent with that given in the Elmer/ice online documentation detailing it's implementation in the model code and notes that as it appears there $K$ is equal to $K/\rho_{ice}$. To avoid confusion, we will use the equation for $K$ as it appears in the supplementary material of Gagliardini et. a (2010). We have also added further text and an equation to describe how this alters the total stress balance of the system, describing how the force contributes to the stress balance. This section now reads:

"A buttressing-like force due to friction along glacier side walls is included through by adding a body force into the force balance using a parameterisation relating lateral resistance to the rheological parameters of the ice and a ice shelf embayment width described by [3]. The body force is given by: $\mathbf{f} = -K|\mathbf{u}|^{m_{lr}-1}\mathbf{u}$

where $m_{lr} = 1/n$ is the lateral resistance exponent where the lateral resistance parameter with $n$ the usual Glen's law parameter. The resistance parameter $K$ is given by: $K = \frac{(n+1)^{1/n}}{W^{\frac{n+1}{n}}(2A)^{1/n}}$ with $A$ the fluidity parameter of the ice. $W$ is a parameter corresponding to a channel half-width which we use to modify the lateral drag during the experiment from being initially high (i.e. low $W$) and then decreased by changing to a high value of $W$ as a means of forcing the glacier to retreat."

*P4 L16: "resulting in a total spinup time of 25000 years" $\rightarrow$ It seems to me that this is not consistent with the description of the spinup procedure given in*

*the previous lines: 16600 years of spinup with 1 m yr 1 of top surface accumulation followed by, at least, 10000 years of spinup with 0.3 m yr 1 of top surface accumulation as you say that: "We determine that the spinup has finished and the ice sheet has reached a steady state when there has been no change in the grounding line position, and the mesh velocity, determining the change in the top and bottom free surfaces, remains less than 0.001 m yr 1 over 10000 years"*
Thank you for pointing out this error, the correct procedure was 10000 years at 1 m yr -1 and then 0.3 m yr -1 held thereafter for 25000 years total. This is now corrected in the text.

*P4 L19: "an" → typo*
Corrected "an" → "and".

*P4 L26: as already said, the way you choose the Weertman law friction (or sliding ?) parameter is not clear to me.*

We have amended the text in the paper to clarify this: "Recently, [2] showed that far from the grounding line, (i.e. for large values of height above flotation) the Weertman and Cavitation sliding relations give an approximately equivalent relationship between basal velocity $u_b$ and basal sheer stress $\tau_b$. For this study we chose a Weertman friction coefficient such that the Weertman and Cavitation relations give similar values of $\tau_b$ far from the grounding line (with high height above flotation) and that would also result in the initial position of the grounding line being within a few km for both sets of experiments."

*P4 L28: you have also run simulations for W = 400 km and W = 500 km based on Fig. 3.*
This is now explicitly stated in the text.

*P5 L11-18: in my opinion, it would make more sense to have this part in the discussion section. In addition, I think that the point discussed here could be better illustrated by a plot of the thickness rates of change $\partial H/\partial t$ as a function of x at different times following buttressing release.*
Since the text is describing what is seen in the plots it refers to, we are happy to leave this in the results section with further discussion about the differences seen between the sliding laws left to the discussion. The thickness rate plot is an interesting idea that we have investigated at your suggestion. While it definitely shows some changes in the thinning rate around the regrounding event, the change in velocity across the grounding line shows this much more clearly. We include the plot (focussing on the regrounding event region and time with the colour scale cropping out larger negative values to see detail). It shows that the thinning increases across the ice shelf after the regrounding event has passed but does not illustrate the temporary slowing in ice velocity across the grounding line in the way that Figure 1c does. In addition, in response to a further comment about Fig. 1c, we have added some discussion about it as detailed below for P6 Fig1.

[Figure]

*P5 L23-26: as already said, this result might be highly sensitive to the value attributed to the C parameter in Eq.(2). Therefore, I would suggest to run similar simulations with higher values of this parameter.*
We agree that the results might be sensitive to the value of C. However, the results may be sensitive to many inputs, including different forcings and geometries. A study of all of them would be interesting but also a much larger undertaking than can be incorporated into revisions of the current paper.

*P6 Fig1: I think you never refer to Fig. 1c in the text, therefore I wonder if it is really relevant to have it here.*
Fig. 1c (and it's counterpart in Fig 3b) is to show the reduction in the ice velocity across the grounding line at the same time that the regrounding occurs. In two-column format these plots are designed to be stacked vertically and the slowdown in the velocity is more pronounced. The paragraph that previously at P6L19 in which Fig1b is discussed now includes discussion of Fig.1c. "The position of the flux gate in Figure 1b (1200km from the ice divide) is chosen as it is located where the regrounding occurs. The flux reaches a maximum as the grounding line approaches the inland end of the retrograde bedrock region, and decreases as the grounding line migrates up the prograde slope. Similarly, we see a reduction in the sliding speed of the ice across the grounding line as shown in Figure 1c. For the cases where dynamic regrounding occurs we see a temporary reduction in the flux and sliding speed, but this reduction is not sufficient to stabilise retreat." We have added further explanation of the velocity plots in the discussion in relation to the qualitative differences seen in the results between the two sliding relations used: "The plots of sliding velocity across the grounding line in Figures 1(c) and 3(b) also point to differences resulting from the choice of sliding relation used. After the initial adjustment in response to the change in buttressing the peak grounding line ice velocity corresponds to the time when the grounding line position changes from lying on reverse bedrock to the positive slope. For the Cavitation sliding case we also see a small dip in velocity when the regrounding occurs. As previously stated we have chosen a Weertman friction coefficient such that both relations give similar basal drag inland (high height above floatation). In the Weertman case the friction must increase towards

the grounding line position (because the ice velocity has increased while the in the Cavitation sliding case, the friction must to decrease due to the effective pressure dependence. This results in the Weertman sliding case showing higher basal friction and slower velocities compared to the Cavitation sliding case."

*P6 L5: "Fig. 3a" I think you mean Fig. 4a*
Yes, this is now corrected to refer to Fig. 4a.

*P7 L6 and P8 L1: "but we cannot rule out the possibility that such behaviour could stabilise a retreating ice sheet for certain geometries." → This is a too strong statement considering the results that are presented in your manuscript for which the transient regrounding obtained with the Schoof law for W = 350 km and W = 375 km are far from preventing the GL to retreat over the retrograde slope.*
While it may be unlikely that this type of dynamic regrounding can fundamentally stabilise an ice sheet, such a possibly cannot be completely ruled out, because there are a lot of competing processes and feedbacks that could kick in. We have altered this sentence now to: "The current study demonstrates temporary regrounding, associated with a small drop in ice flow velocities. Whether or not this kind of dynamic regrounding could occur on larger spatial or temporal scales, or even stabilise a marine ice sheet, cannot be inferred from the current study."

*P8 L9: "ice shelf ice shelf" → typo*
Corrected.

*P8 L13: "force balance" → In my opinion, "stress distribution" or "stress state" would be more appropriate*
We have changed "force balance" to "stress state" as per the reviewers request.

*P8 L20: "transition zone" → I don't agree with the use you make of the term "transition zone" in this case. In line with Pattyn and others (2006), I understand the "transition zone" as being the narrow region right upstream the GL over which $\tau_b$ progressively vanishes. By construction, the Weertman law with a uniform friction coefficient leads to a discontinuity of $\tau_b$ at the GL which is equivalent to say that the length of the transition zone is reduced to 0. There cannot be any "transition zone" within the ice shelf - as you seem to suggest - as $\tau_b = 0$ wherever ice is floating.*

To clarify our meaning here, we have changed this existing wording: "For the case of Weertman sliding, the instantaneous decay of basal shear stress from its maximum value to zero as the grounding line is crossed (Figure 4) causes the transition zone to extend into the ice shelf several km (typically around 20km in our experiments) causing the concave and rapidly thinning ice shelf in this region." to the following: "For the case of Weertman sliding, the basal shear stress drops from its maximum value to zero as the grounding line is crossed (Figure 4). The high basal shear stress right up to the grounding line is balanced by a correspondingly high driving stress, with maximum surface slope occurring at the grounding line. Thus instead of a transition zone upstream

of the grounding line, a geometrically concave region with very high spatial gradients in driving stress and flow speed extends downstream into the ice shelf (typically around 20km in our experiments) before a more typical shelf regime is attained."

*P9 L5: this result was already highlighted in Tsai and others (2015), Gladstone and others (2017) and Brondex and others (2017) (the Budd law being investigated in the two latter), and therefore a citation of these studies would be welcome here.*
Yes, our results are consistent with these results and citations have been added. The final line of this paragraph now reads: "This difference in ice shelf profile is a direct result of dependence on effective pressure at the bed and is likely to be present also for other sliding relations featuring such a dependence, as has been shown in (e.g. Tsai 2015, Gladstone 2017,Brondex 2017)"

*P10 L5-6: "in which regrounding of an ice shelf after retreat has stabilised may occur through bedrock uplift after ice unloading"* → *I don't understand the meaning of this sentence, is there a problem with it ?*
The text has been corrected to read "in which regrounding of an ice shelf after grounding line retreat may occur through bedrock uplift after ice unloading."

*P10 L7: to me, it is not very clear to what timescale you refer. Is it the duration of the regrounding ?*
The duration of regrounding following this overshoot process can be more or less permanent, so the timescale here refers to the whole process from the onset of retreat to when the regrounding occurs and is of order hundreds of years or thousands of years. To clarify we have changed "timescale for dynamic regrounding" to "timescale for dynamic regrounding in the current study", because timescales may well be different for different systems and would likely depend on bedrock geometry.

*P11 L13: "For each channel width" This formulation is misleading as the cases W = 250 km , W = 375 km and W = 450 km do not seem to be tested in your sensitivity analysis.*
Text now explicitly states "For values of W=300km, 350km and 400km"

*P11 L14: "We conclude that the effects shown here are not dependent on the mesh resolution" here too the formulation is misleading: if I am correct (based on Fig. 6), the 2 km mesh spacing case shows no regrounding at all even for W = 350 km so it is not correct to state that your results are not dependent on the mesh resolution.*
By this we mean that we are confident that we have enough resolution to show that the effect is not simply an artefact of mesh resolution, showing that the results for the two finer resolutions using 1km and 500m mesh spacings are consistent. To clarify we have altered the text to read : "We conclude that, for resolutions of 1km or finer, the effects shown here are not dependent on the mesh resolution"

*P12 L3: "Previous studies into the mesh resolution" → I am not a native english speaker but this formulation sounds odd to me"*
*P3P12 L4: e.g. (Durand et al., 2009)" → (e.g. Durand et al., 2009)*
We have corrected this sentence for clarity to: "Previous studies of mesh resolution dependence of the grounding line position and evolution using Weertman sliding, e.g. Durand (2009), have demonstrated convergence of the grounding line position."

*P13: you need to correct the friction law parameters after having rewritten the equations of P3. In addition, m should be 1/3 and not 3.*
The equations on P3 and the parameters have been corrected accordingly in this table.

**References**

[1] G. Durand, O. Gagliardini, B. de Fleurian, T. Zwinger, and E. Le Meur. Marine ice sheet dynamics: Hysteresis and neutral equilibrium. *Journal of Geophysical Research*, 114(F3):F03009, aug 2009.

[2] Julien Brondex, Olivier Gagliardini, Fabien Gillet-Chaulet, and Gaël Durand. Sensitivity of grounding line dynamics to the choice of the friction law. *J. Glaciol.*, 63(241):854–866, oct 2017.

[3] O. Gagliardini, G. Durand, T. Zwinger, R. C. A. Hindmarsh, and E. Le Meur. Coupling of ice-shelf melting and buttressing is a key process in ice-sheets dynamics. *Geophysical Research Letters*, 37(14):n/a–n/a, jul 2010.

---

## Referee Report (RR1)

**General comments**

Overall, the authors of "Simulated dynamic regrounding during marine ice sheet retreat" have taken the comments made by Dr Todd and myself into account and the quality of the manuscript has improved. However, there are still some points that I don't really share with the authors and which are listed below. Then, in a second section, I adress a few specific issues that are left in the new version of the manuscript.

First of all, there is still a problem with your corrected version of Eq. (3). If you want your notations to follow the ones used in Brondex and others (2017) as stated in your response to reviewers comments then it should be:

$$\chi = \frac{u_b C_S^n}{C_{max}^n N^n} \tag{1}$$

The reason for that is that the relationship between the factor $A_S$ you used in the first version of your manuscript and the factor $C_S$ that has been used by Brondex and others (2017) is $A_S = C_S^{-n}$. Indeed, by construction, the cavitation friction law is such that it is perfectly equivalent to a Weertman-type friction law (i.e. $\tau_b \to C_S u_b^m$) far upstream the GL where $N$ is very high and perfectly equivalent to a Coulomb-type friction law (i.e. $\tau_b \to C_{max} N$) at the GL where $N \to 0$ ; if you were to keep Eq. (3) as it is in the current version of the manuscrit then there would be an inconsistency between the $C_W$ of Eq. (5) and the $C_S$ of Eq. (3) and, in addition, the units given for $C_S$ in Table 1 would be wrong.

This lead me to the heart of my criticism: the way you chose the values of the friction parameters $C_W$ and $C_S$ (which you still describe as a "sliding parameter" in P3 L16 while it should be called a "friction parameter" since an increase of $C_S$ leads to an increase of $\tau_b$) to get your two initial states is still unclear. In the new version of the text, it is said that "For this study we chose a Weertman friction coefficient such that the Weertman and Cavitation relations give similar values of $\tau_b$ far from the grounding line (with high height above flotation) and that would also result in the initial position of the grounding line being within a few km for both sets of experiments". But in that case, given the aforementioned asymptotic behavior of the cavitation friction law at high effective pressure, $C_W$ and $C_S$ should have the same value which is not the case looking at Table 1. Is that a consequence of the mistake you made in Eq. (3) ? I don't ask you to follow the same initialisation procedure as in Brondex and others (2017) but there is definitely something which needs to be clarified here.

The second point I wanted to raise regards the sensitivity of your results to the value attributed to the parameter $C_{max}$. It is stated in your response that "[you] don't see an argument why $C_{max}$ should be especially important". By constrcution, the value of the parameter $C_{max}$ controls the length of the region located upstream the GL over which the cavitation friction law reduces to a Coulomb law: the higher this value, the narrower this region. In other word, if the value attributed to $C_{max}$ would have been higher than the value you have chosen (i.e. $C_{max} = 0.1$), the values of $\tau_b$ given by the two friction laws would have been closer and, as a consequence, the bottom ice shelf profile obtained with the cavitation friction law right downstream the GL would probably have been more similar to the strongly concave shape you get with the Weertman law (Brondex and others, 2017, see). Yet, you do admit in your manuscript that "The strongly concave lower surface of the ice shelf just downstream from the grounding line in the case of Weertman sliding increases the water column depth under the ice shelf and reduces the likelihood of regrounding." That is precisely the reason why I do think that higher values of $C_{max}$ could have prevented the "dynamic regrounding" to occur with the cavitation friction law. Although it would have been the best thing to do, I agree that a sensitivity study is a large undertaken and I don't expect you to perform one. However, to my mind it would be good to state in your paper that the ice shelf bottom profile obtained with the cavitation law might be sensitive to the value attributed to the parameter $C_{max}$ with higher values giving a more concave shape and, as such, reducing the likelihood of dynamic regrounding.

**Specific comments**

First of all, note that the pages and lines listed below refer to the version of the manuscript with the blue and red colors which follows the authors response to the reviewers comments in the same document.

P2 L3: "such as Pine Island Glacier, have been also been a particular focus " $\to$ typo

P2 L7: "Brondex et al. (2017)" $\to$ (Brondex et al., 2017)

P2 L10: I think you mean "shear" and not "sheer", don't you ?

P2 L14: which are implemented ?

P2 L17: You need to reformulate this sentence as the Budd law does not satisfy the Iken bound (the Schoof law does).

P3 L13: Eq (3) needs to be corrected for consistency with Eq. (5). See general comments.

P3 L16: $C_S$ is a friction parameter. In addition, it is still not stated in the text that in your case $q = 1$ and why this value has been chosen.

P4 L1: I think there is a problem with this sentence.

P5 L7: See general comments.

P5 L22: Continues to retreat ?

P8 L9-10: There are several problems in this sentence.

P8 L14 to P9 L1: The first and second sentences of the discussion section give redundant information. I think that they should be reformulated.

P11 L3: Remove the quote mark at the end of the line.

P11 L9: Some parenthesis need to be removed here.

P11 L10-17: There are many things that should be changed in this paragraph. First of all, there are several english mistakes. Second, the sentence starting at L11 is not clear at all to me. Also, I don't understand why there is, from time to time, a capital C at the beginning of the word Cavitation (it is also the case elsewhere in the manuscript). I advice you to read it again carefully and correct it.

P12 L17: I have the feeling that this sentence and the one L14P12 contradict each other. At least, the fact that a 250m mesh resolution at the GL is sufficient to prevent numerical arctefact when using a Weertman friction law is not really convincing from what is written in this paragraph.

Table 1: $C_W$ and $C_S$ ought to be called friction parameters instead of sliding parameters as an increase of their values leads to an increase of $\tau_b$

**References**

Brondex J, Gagliardini O, Gillet-Chaulet F and Durand G (2017) Sensitivity of grounding line dynamics to the choice of the friction law. Journal of Glaciology, **63**(241), 854–866

---

## Author Response (AR2)

**Response to general comments from the reviewer**

We would like to thank the reviewer for their careful reading and insightful comments. In an effort to clarify the parameters for the two sliding relations used a mistake was made that introduced an inconsistency between equation 3 and the value stated in table 1 for $C_S$. Careful examination of the values used in the simulations and the equations in question has shown that the value quoted in table 1 is indeed the value of $A_S$, the sliding parameter, not a friction coefficient. The values quoted are now the parameters and the values used in the actual simulations (with $A_S$, and $C_W$ being the parameters used by the built-in functions in Elmer/Ice to specify the cavitation and Weertman sliding laws respectively, consistent the with notation used in the Elmer/Ice wiki and Gagliardini et al. (2007). In the previous revised version of the manuscript we had used the notation to notation of Brondex et al (20017) and the friction coefficient $C_S$, we feel it is clearer to use the notation from the original manuscript which better reflects the work that was done. The values of $A_S$ and $C_W$ that were used were not selected for the sliding relations to be exactly equivalent, but to result in an ice shelf with similar grounding line positions. We hope this answers the reviewers concerns about the experimental setup and clarifies our choice of coefficients.

We agree with the reviewer that the sliding velocity and hence shape of the ice shelf would indeed be sensitive to the choice of all the model parameters used, including $C_{\max}$. Our value of 0.1 is a reasonable choice of for this. It is not however the only control on the resultant shape of the ice shelf. That said, a sensitivity study of the model to a range of the parameters including those of the sliding relation and the bedrock geometry would be a valuable future research project. We have included the following text at the end of the third paragraph in the discussion: "For higher values of $C_{\max}$ the transition zone would be smaller, and the ice sheet geometry closer to that obtained with Weertman sliding. Thus the occurrence of dynamic regrounding may depend not only on choice of sliding relation but also parameter choices for the chosen sliding relation."

**Response to specific comments**

We would like to thank the referee for their careful reading, in most cases we have simply followed their suggestions and these changes can be seen in the diff of the manuscript. Where more detailed response was warranted please see below.

*P3 L16: $C_S$ is a friction parameter. In addition, it is still not stated in the text that in your case q = 1 and why this value has been chosen.*
Please see above in the response to general comments about the friction parameter. We also now state in the text that $q = 1$. This value is consistent with similar studies which set $q = 1$.

*P8 L9-10: There are several problems in this sentence.*
This sentence now reads: "we see high frequency changes in velocity, with each

jump in velocity likely to correspond to grounding line movement between neighbouring mesh elements"

*P8 L14 to P9 L1: The First and second sentences of the discussion section give redundant information. I think that they should be reformulated.*
These two sentences have now been combined to read: "Retreat simulations in the current study have demonstrated that regrounding of an ice shelf, associated with a small drop in ice flow velocities, may occur under certain conditions during the rapid, unstable retreat of a marine ice sheet."

*P11 L10-17: There are many things that should be changed in this paragraph. First of all, there are several english mistakes. Second, the sentence starting at L11 is not clear at all to me. Also, I don't understand why there is, from time to time, a capital C at the beginning of the word Cavitation (it is also the case elsewhere in the manuscript). I advice you to read it again carefully and correct it.*
This paragraph has been edited for clarity and to correct the captilisation of "Cavitation".

"The plots of sliding velocity across the grounding line in Figures 1c and 3b also point to differences resulting from the choice of sliding relation. After the initial adjustment period in response to the change in buttressing, the peak in grounding line ice velocity corresponds to the time when the position of the grounding line crosses from being situation on a retrograde to a prograde bedrock slope. In the cavitation sliding case we also see a small drop in velocity occurring at the same time as the regrounding. Moving from inland towards the grounding line the value of basal friction changes markedly depending on the sliding relation used. In the Weertman case the basal friction must increase towards the grounding line position (because the ice velocity has increased) while in the cavitation sliding case, the friction must decrease due to the effective pressure dependence. This results in the Weertman sliding case showing higher basal friction and slower velocities compared to the cavitation sliding case."

*P12 L17: I have the feeling that this sentence and the one L14P12 contradict each other. At least, the fact that a 250m mesh resolution at the GL is suffcient to prevent numerical arctefact when using a Weertman friction law is not really convincing from what is written in this paragraph.*
We have altered this paragraph to the text below. Further experiments using even finer mesh resolutions would be computationally expensive, but both the characteristic ice shelf shape preventing regrounding and the final grounding line position would still be consistent.

"Similar experiments were performed using the Weertman sliding relation on uniform meshes with 1km, 500m and 250m grid spacing. In this case, while the end position of the grounding line appears to converge, the finer mesh results in faster retreat of the grounding line across the retrograde slope. The concave geometry of the lower surface of the ice shelf, however, is consistent across the mesh resolution experiments. Previous studies into the mesh resolution dependence of the grounding line position and evolution using Weertman sliding, eg. [3], have shown that consistency in the final grounding line positions can be obtained with horizontal mesh elements of below 5km."

*Table 1: $C_W$ and $C_S$ ought to be called friction parameters instead of sliding parameters as an increase of their values leads to an increase of $\tau_b$*

Corrected to $C_W$ to be labelled friction parameter, the cavitation sliding law is now back to the previously used notation, with $A_S$ labelled as the "sliding parameter" with corrected units.

[revised manuscript text omitted]